# CityGaussianV2: Efficient and Geometrically Accurate Reconstruction for Large-Scale Scenes

**Yang Liu[1,2], Chuanchen Luo[3], Zhongkai Mao[1,2], Junran Peng[4] ✉, & Zhaoxiang Zhang [1,2] ✉**
[1] NLPR, MAIS, Institute of Automation, Chinese Academy of Sciences
[2] University of Chinese Academy of Sciences
[3] Shandong University     [4] University of Science and Technology Beijing
{liuyang2022, maozhongkai2023, zhaoxiang.zhang}@ia.ac.cn
chuanchen.luo@sdu.edu.cn,  jrpeng4ever@126.com

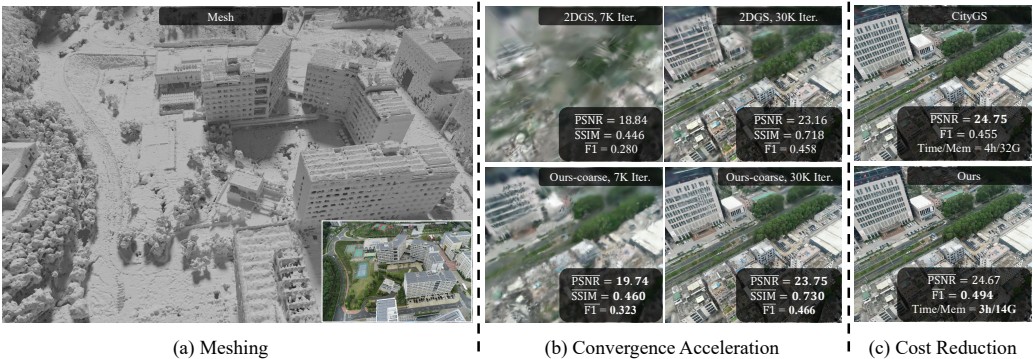

Figure 1: Illustration of the superiority of CityGaussianV2. (a) Our method reconstructs large-scale complex scenes with accurate geometry from multi-view RGB images, restoring intricate structures of woods, buildings, and roads. (b) "Ours-coarse" denotes training 2DGS with our optimization algorithm. This strategy accelerates 2DGS reconstruction in terms of both rendering quality (PSNR, SSIM) and geometry accuracy (F1 score). (c) Our optimized parallel training pipeline reduces the training time and memory by 25% and 50% respectively, while achieving better geometric quality. We report mean quality metrics in *GauU-Scene* (Xiong et al., 2024) here, with the best performance in each column highlighted in **bold**.

## Abstract

Recently, 3D Gaussian Splatting (3DGS) has revolutionized radiance field reconstruction, manifesting efficient and high-fidelity novel view synthesis. However, accurately representing surfaces, especially in large and complex scenarios, remains a significant challenge due to the unstructured nature of 3DGS. In this paper, we present CityGaussianV2, a novel approach for large-scale scene reconstruction that addresses critical challenges related to geometric accuracy and efficiency. Building on the favorable generalization capabilities of 2D Gaussian Splatting (2DGS), we address its convergence and scalability issues. Specifically, we implement a decomposed-gradient-based densification and depth regression technique to eliminate blurry artifacts and accelerate convergence. To scale up, we introduce an elongation filter that mitigates Gaussian count explosion caused by 2DGS degeneration. Furthermore, we optimize the CityGaussian pipeline for parallel training, achieving up to $10\times$ compression, at least 25% savings in training time, and a 50% decrease in memory usage. We also established standard geometry benchmarks under large-scale scenes. Experimental results demonstrate that our method strikes a promising balance between visual quality, geometric accuracy, as well as storage and training costs. More live demos and official code implementation are available at our project page: https://dekuliutesla.github.io/CityGaussianV2/.

# 1 INTRODUCTION

3D scene reconstruction is a long-standing topic in computer vision and graphics, with its core pursuit of photo-realistic rendering and accurate geometry reconstruction. Beyond Neural Radiance Fields (NeRF) (Mildenhall et al., 2021), 3D Gaussian Splatting (3DGS) (Kerbl et al., 2023) has become the predominant technique in this area due to its superiority in training convergence and rendering efficiency. 3DGS represents the scene with a set of discrete Gaussian ellipsoids and renders with a highly optimized rasterizer. However, such primitives take an unordered structure and do not correspond well to the actual surface of the scene. This limitation impairs its synthesis quality at extrapolated views and hinders its downstream application in editing, animation, and relighting (Guédon & Lepetit, 2024). Recently, many excellent works (Guédon & Lepetit, 2024; Huang et al., 2024; Yu et al., 2024c) have been proposed to address this issue. Despite their great success in single objects or small scenes, devils emerge when applying them directly to complex, large-scale scenes.

On the one hand, existing methods face significant challenges related to scalability and generalization ability. For example, SuGaR (Guédon & Lepetit, 2024) binds meshes with Gaussians for refinement. However, it struggles to recover complex geometry details (Fig. 6) and can trigger out-of-memory errors when scaling up due to suboptimal implementation. GOF (Yu et al., 2024c) struggles with large, over-blurred Gaussians. These Gaussians obstruct the field of view and hinder valid supervision, leading to severe underfitting and shell-like mesh that is non-trivial to remove, as validated in Fig. 7 and Fig. 8 in Appendix. While 2DGS (Huang et al., 2024) exhibits better generalization ability, as shown in Tab. 1, its convergence is hindered by the blurred Gaussians illustrated in part (b) of Fig. 1. Additionally, when scaling up through parallel training, it suffers from a Gaussian count explosion, as depicted in Fig. 3. Another challenge lies in the evaluation protocol: due to insufficient observations in boundary regions, geometry estimation becomes error-prone and unstable in these areas. As a result, the metrics can significantly fluctuate and underestimate actual performance (Xiong et al., 2024), making it difficult to objectively evaluate and compare algorithms.

On the other hand, achieving efficient parallel training and compression is critical to realizing geometrically accurate reconstruction of large-scale scenes. The total number of Gaussians can increase to 19.3 million during parallel training, resulting in a storage requirement of 4.6 GB and a memory cost of 31.5 GB, while rendering speed drops below 25 FPS. Additionally, existing VastGaussian (Lin et al., 2024) costs nearly 3 hours for training, and CityGaussian Liu et al. (2024) consumes 4 hours to finish both training and compression. For reconstruction on low-end devices or under strict time constraints, these training costs and rendering speeds are unacceptable. Therefore, there is an urgent need for an economical parallel training and compression strategy.

In response to these challenges, we introduce CityGaussianV2, a geometrically accurate yet efficient strategy for large-scale scene reconstruction. We take 2DGS as primitive due to its favorable generalization capabilities. To accelerate reconstruction, we employ depth regression guided by Depth-Anything V2 (Yang et al., 2024) and Decomposed-Gradient-based Densification (DGD). As shown in part (b) of Fig. 1 and Tab. 2, DGD effectively eliminates blurred surfels, crucial for performance improvement. To address scalability, we introduce an Elongation Filter to mitigate the Gaussian count explosion problem associated with 2DGS degeneration during parallel training. To reduce the burden of single GPU, we conduct parallel training based on CityGaussian's block partitioning strategy. And we streamline the process by omitting time-consuming post-pruning and distillation steps of CityGaussian. Instead, we implement spherical harmonics of degree 2 from scratch and integrate contribution-based pruning into per-block fine-tuning. As demonstrated in part (c) of Fig. 1, it scales up the surface quality of complex structures while significantly reducing training costs. Furthermore, our contribution-based vectree quantization enables a tenfold reduction in storage requirements for large-scale 2DGS. For evaluation, we introduce TnT-style (Knapitsch et al., 2017) protocol along with a visibility-based crop volume estimation strategy, which can efficiently exclude underobserved regions and bring stable and consistent assessment.

In summary, our contributions are four-fold:

- A novel optimization strategy for 2DGS, that accelerates its convergence under large-scale scenes and enables it to be scaled up to high capacity (Sec. 3.2).

- A highly optimized parallel training pipeline that significantly reduces training costs and storage requirements while enabling real-time rendering performance (Sec. 3.3).

- A TnT-style standardized evaluation protocol tailored for large, unbounded scenes, establishing a geometric benchmark for large-scale scene reconstruction (Sec. 4).
- To the best of our knowledge, our CityGaussianV2 is among the first to implement the Gaussian radiance field in large-scale surface reconstruction. Experimental results confirm our state-of-the-art performance in both geometric quality and efficiency.

## 2 RELATED WORKS

### 2.1 NOVEL VIEW SYNTHESIS

**Novel view synthesis** aims at generating new images from previously unseen viewpoints using images captured from various source viewpoints around a 3D scene. These new renderings are primarily based on the reconstructed 3D representation of the scene. One of the most seminal contributions to this field is **Neural Radiance Fields (NeRF)** (Mildenhall et al., 2021), which implicitly models target scenes using multi-layer perceptions (MLPs). Following this, MipNeRF (Barron et al., 2021; 2022) addresses objectionable aliasing artifacts by introducing anti-aliased conical frustum-based rendering. Deng et al. (2022); Wei et al. (2021); Xu et al. (2022) apply depth supervision from point cloud to accelerate model convergence. Algorithms represented by InstantNGP (Müller et al., 2022) speeds up the training and rendering of NeRF by leveraging simplified data structures, including multi-resolution hash encoding grid and octrees (Zhang et al., 2023; Wang et al., 2022; Yu et al., 2021). The recently emerging **3D Gaussian Splatting** (Kerbl et al., 2023) overcomes NeRF's drawbacks in training efficiency and rendering speed. Follow-up works further improve upon 3DGS in anti-aliasing Yu et al. (2024b), storage cost Fan et al. (2023); Zhang et al. (2024c); Navaneet et al. (2023); Morgenstern et al. (2023), and high-texture area underfitting Bulò et al. (2024); Zhang et al. (2024b). These remarkable works have provided valuable insights into the design of our algorithm.

### 2.2 SURFACE RECONSTRUCTION WITH GAUSSIANS

Extracting accurate surfaces from unordered and discrete 3DGS is a challenging while intriguing task. A handful of algorithms have been developed to extract unambiguous surfaces and regularize smoothness and outliers. Pioneering SuGaR (Guédon & Lepetit, 2024) pretrain 3DGS and bind it with extracted mesh for fine-tuning. It then relies on Poisson reconstruction algorithm for fast mesh extraction. Recent GSDF (Yu et al., 2024a) and NeuSG (Chen et al., 2023) optimize 3DGS together with a signed distance function to generate accurate surfaces. 2DGS (Huang et al., 2024) and concurrent GaussianSurfels (Dai et al., 2024) collapse one dimension of 3D Gaussian primitives to avoid ambiguous depth estimation. The normals derived from rendering and depth map are also aligned to ensure a smooth surface. TrimGS (Fan et al., 2024) further provides a novel per-Gaussian contribution definition to remove inaccurate geometry. As a post-processing technique, GS2Mesh (Wolf et al., 2024) uses a pre-trained stereo-matching model to export mesh from 3DGS directly. GOF (Yu et al., 2024c) focuses on unbounded scene. It leverages ray-tracing-based volume rendering to obtain contiguous opacity distribution within the scene. Instead of 2DGS's TSDF-based marching-cube strategy, GOF gets SDF from the opacity field and use marching tetrahedra to extract mesh. RaDeGS (Zhang et al., 2024a) novelly define the ray intersection with Gaussian and correspondingly derive curved surface and depth distribution. Though these algorithms have been proven to be successful on small scenes or single objects, the challenges behind scaling up, including performance degradation, densification stability, and training cost, remain unexplored. We hope our analysis and design can provide more insights into the community.

### 2.3 LARGE-SCALE SCENE RECONSTRUCTION

Over the past few decades, 3D reconstruction from large image collections has gained considerable attention and made significant strides. Modern algorithms (Tancik et al., 2022; Turki et al., 2022; Xiangli et al., 2022; Xu et al., 2023; Zhang et al., 2023; Li et al., 2024) are largely based on NeRF (Mildenhall et al., 2021). However, the substantial time required for training and rendering has hindered NeRF-based methods for long time. The recent rise of 3DGS, exemplified by VastGaussian (Lin et al., 2024), represents a paradigm shift in large-scale scene reconstruction. Subsequent developments like HierarchicalGS (Kerbl et al., 2024) and OctreeGS (Ren et al., 2024) have introduced Level-of-Detail (LoD) techniques, enabling efficient rendering of scenes at various scales. CityGS

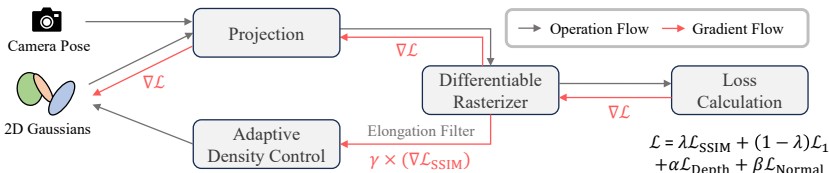

Figure 2: Illustration of our optimization mechanism. We densify Gaussians exclusively according to the gradient of SSIM loss. This helps remove large and blurry Gaussians and accelerate convergence. Meanwhile, we disable the densification of Gaussians with extreme elongation to avoid the Gaussian count explosion shown in Fig. 3. We also supervise the rendered depth with that predicted by Depth Anything V2 (Yang et al., 2024). This helps improve both rendering and geometry quality.

(Liu et al., 2024) presents a comprehensive pipeline that encompasses parallel training, compression, and LoD-based fast rendering. And DoGaussian (Chen & Lee, 2024) applies Alternating Direction Methods of Multipliers (ADMM) to train 3DGS distributedly. Meanwhile, GrendelGS (Zhao et al., 2024) facilitates communication between blocks on different GPUs, and FlashGS (Feng et al., 2024) significantly reduces VRAM costs for large-scale training and rendering through a highly optimized renderer. Despite these advances, the issue of geometry accuracy has been largely overlooked due to the lack of reliable benchmarks. Our work addresses this gap, proposing a reliable benchmark along with a novel algorithm for both economical training, high fidelity, and accurate geometry.

## 3 METHOD

### 3.1 PRELIMINARY

**3D Gaussian Splatting** (Kerbl et al., 2023) represents 3D scene with a set of ellipsoids described by 3D Gaussian distribution, i.e. $\mathbf{G_N} = \{G_n | n = 1, ..., N\}$. Each Gaussian contains learnable properties including central point $\boldsymbol{\mu_n} \in \mathbb{R}^{3 \times 1}$, covariance $\boldsymbol{\Sigma_n} \in \mathbb{R}^{3 \times 3}$, opacity $\sigma_n \in [0, 1]$, spherical harmonics (SH) features $\boldsymbol{f_n} \in \mathbb{R}^{3 \times 16}$ for view-dependent rendering. The covariance matrix is further decomposed to scaling matrix $\mathbf{S_n}$ and rotation matrix $\mathbf{R_n}$, i.e. $\boldsymbol{\Sigma_n} = \mathbf{R_n} \mathbf{S_n} \mathbf{S_n}^T \mathbf{R_n}^T$. For a certain pixel $p$, the color $\boldsymbol{c}_p$ is derived through alpha blending:

$$\boldsymbol{c}_p = \sum_{i \in \gamma(p)} \boldsymbol{c}_i \alpha_i \prod_{j=1}^{i-1} (1 - \alpha_j),$$

$$\alpha_i = \sigma_i \cdot \exp\left(-\frac{1}{2} (\boldsymbol{x} - \boldsymbol{\mu}_i)^T \boldsymbol{\Sigma}_i^{-1} (\boldsymbol{x} - \boldsymbol{\mu}_i)\right),$$

(1)

where $\gamma(p)$ denotes Gaussians located on ray crossing pixel $p$, and $\boldsymbol{x}$ is the corresponding query point. The loss $\mathcal{L}$ that supervises 3DGS's optimization is the weighted sum of two parts, L1 loss $\mathcal{L}_1$ and D-SSIM loss $\mathcal{L}_{\text{SSIM}}$. 3DGS prevents under or over-reconstruction through heuristic adaptive density control, which is guided by view-space position gradient, i.e. $\nabla_{densify} = \partial \mathcal{L} / \partial \boldsymbol{\mu}_n$. The Gaussians with a gradient larger than a certain threshold would be cloned or split. For more details, we refer the readers to the original paper of 3DGS (Kerbl et al., 2023).

**CityGaussian** (Liu et al., 2024) aims to scale up 3DGS to large-scale scenes. As shown in Fig. 4, it first pre-trains a coarse model on full training data with the schedule of 3DGS. After that, it divides Gaussian primitives and training data into non-overlapping blocks and conducts parallel tuning. Following this, it adopts the approach from LightGaussian (Fan et al., 2023), applying an additional 30,000 iterations for pruning and 10,000 iterations for distillation. Pruning removes redundant Gaussians based on their rendering importance, while the distillation reduces the spherical harmonic (SH) degree from 3 to 2. It then conducts vectree quantization for storage compression.

**2D Gaussian Splatting** (Huang et al., 2024) addresses surface estimation ambiguity of 3DGS by collapsing 3D ellipsoid volumes into a set of 2D oriented Gaussian disks, known as surfels. Its covariance is characterized by two tangential vectors $\boldsymbol{t}_{n,u}$ and $\boldsymbol{t}_{n,v}$ and a scaling vector $\mathbf{S_n} = (s_{n,u}, s_{n,v})$. In addition, 2DGS incorporates depth distortion regularization and applies sur-

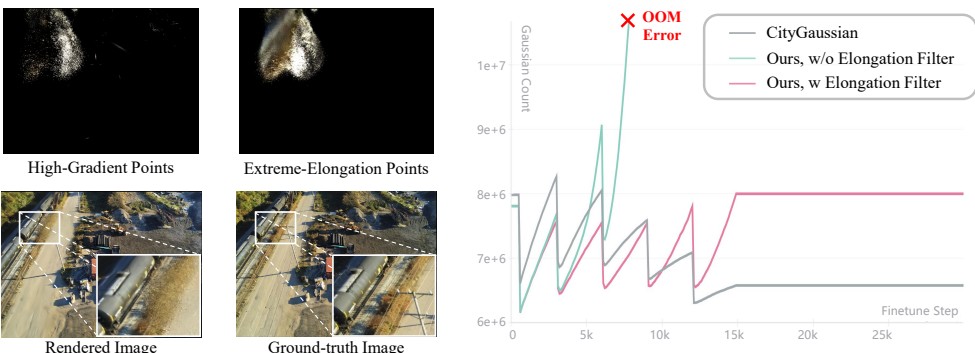

Figure 3: Illustration of the motivation and effectiveness of our Elongation Filter. We take the **tuning** of one block of *Rubble* (Turki et al., 2022) scene as an example. On the left, we highlight the collection of Gaussian primitives with high gradient or extreme elongation. There is a significant overlap between two collections. By restricting densification of these sand-like points, we prevent out-of-memory (OOM) errors caused by an explosion in Gaussian count, enabling a steady count evolution analogous to CityGaussian (Liu et al., 2024) in parallel tuning, as depicted on the right.

face smoothness loss $\mathcal{L}_{\text{Normal}}$ to align the surfel normals with those estimated from the depth map. These enhancements lead to superior results in geometry reconstruction and novel view synthesis.

## 3.2 OPTIMIZATION MECHANISM

This section elaborates on the proposed optimization mechanism for convergence acceleration and stable training. As illustrated in Fig. 2, the mechanism comprises three components: Depth Supervision, Elongation Filter, and Decomposed-Gradient-based Densification (DGD).

As depicted in Fig. 2, 2D Gaussians are projected into screen space at the given camera pose and rendered by a tailored rasterizer. The derived outputs are used for loss calculation. GS algorithm necessitates iterative optimization to disambiguate monocular cues from each view, ultimately converging to a coherent 3D geometry. To encourage convergence, we incorporate depth prior as an auxiliary guidance for geometry optimization. Following the practice in Kerbl et al. (2024), we utilize Depth-Anything-V2 to estimate the inverse depth and align it to the dataset's scale, which we denote as $D_k$. Suppose $\hat{D}_k$ denotes the predicted inverse depth. The associated loss function is defined as $\mathcal{L}_{\text{Depth}} = |\hat{D}_k - D_k|$. As the training progresses, we decrease the loss weight $\alpha$ exponentially to suppress the adverse effect of imperfect depth estimation gradually.

As discussed in Sec. 1, the critical obstacle to scaling up 2DGS is the excessive proliferation of certain primitives during the parallel tuning stage. Typically, a 2D Gaussian can collapse to a very small point when projected from a distance, especially those exhibiting extreme elongation (Huang et al., 2024). With high opacity, the movement of these minuscule points can cause significant pixel changes in complex scenes, leading to pronounced position gradients. As evidenced in the left portion of Fig. 3, these tiny, sand-like projected points contribute substantially to points with high gradients. And they belong to those with extreme elongation. Moreover, some points project smaller than one pixel, resulting in their covariance being replaced by a fixed value through the antialiased low-pass filter. Consequently, these points cannot properly adjust their scaling and rotation with valid gradients. In block-wise parallel tuning, the views assigned to each block are much less than the total. These distant views are therefore frequently observed, causing the gradients of degenerated points to accumulate rapidly. These points consequently trigger exponential increases in Gaussian count and ultimately lead to out-of-memory errors, as demonstrated in the right portion of Fig. 3.

In light of this observation, we implement a straightforward yet effective Elongation Filter to address this problem. Before densification, we assess the elongation rate of each surfel, defined as $\eta_n = \min(s_{n,u}, s_{n,v})/\max(s_{n,u}, s_{n,v})$. Surfels with $\eta_n$ below a certain threshold are excluded from the cloning and splitting process. As shown in the right portion of Fig. 3, this filter mitigates out-of-memory errors and facilitates a more steady Gaussian count evolution. Furthermore, experimental results in Tab. 2 demonstrate that it does not compromise performance at the pretraining stage.

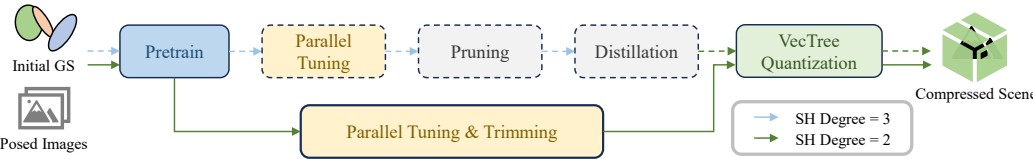

Figure 4: Illustration of pipeline modification. The pipeline of CityGS (Liu et al., 2024) (dashed boxes and arrows) is compared with ours. We successfully removed time-consuming post-pruning and distillation, while enabling storage compression for 2DGS.

Naive 2DGS also suffers from suboptimal optimization when migrated to large-scale scenes. We empirically found that 2DGS is more susceptible to blurry reconstruction than 3DGS at the early training stage, as shown in Fig. 10 of the Appendix. As indicated by Wang et al. (2004); Zhang et al. (2024b); Shi et al. (2024), in contrast to SSIM loss, the L1 RGB loss is insensitive to blurriness and does not prioritize preserving structural integrity. Tab. 7 of the Appendix further ablates on the gradient source of adaptive density control, validating that participation of its gradient is the most critical for sub-optimal results. To alleviate this problem, we prioritize the gradient from SSIM loss and introduce a Decomposed-Gradient-based Densification (DGD) strategy. Specifically, the gradient for densification is reformulated as:

$$\nabla_{densify} = \max\left(\omega \times \frac{|\nabla\mathcal{L}|_{avg}}{|\nabla\mathcal{L}_{\text{SSIM}}|_{avg}}, 1\right) \times \nabla\mathcal{L}_{\text{SSIM}}, \quad (2)$$

where $\nabla\mathcal{L}_{\text{SSIM}}$ is scaled according to the average gradient norm of the total loss to align automatically with the original gradient threshold for densification, with $\omega$ representing a constant weight.

## 3.3 PARALLEL TRAINING PIPELINE

As discussed in Sec. 1, the post-pruning and distillation of CityGaussian leads to time and memory overhead. To resolve these issues, we propose a novel pipeline, as shown in Fig. 4. To bypass the distillation step, we use an SH degree of 2 from the start, reducing the SH feature dimension from 48 to 27. This results in considerable memory and storage savings throughout the whole pipeline. To eliminate the need for post-pruning, we incorporate trimming during block-wise tuning. Specifically, we define the single-view contribution of each Gaussian following Fan et al. (2024):

$$\mathbf{C}_{n,k} = \frac{1}{|\mathbb{P}_k|} \sum_{p \in \mathbb{P}_k} (\alpha_n)^\gamma \left(\prod_{j=1}^{n(p)-1} (1 - \alpha_j)\right)^{(1-\gamma)}, \quad (3)$$

where $\mathbb{P}_k$ is the 2D projected region of $n$-th Gaussian under $k$-th view. $n(p)$ denotes its depth sorted order on ray crossing pixel $p$. $\gamma$ is set as the default value of 0.5. Suppose that the images assigned to $m$-th block using CityGS (Liu et al., 2024)'s strategy is $\mathbb{V}_m$, then the average contribution is:

$$\mathbf{C}_n = \frac{1}{|\mathbb{V}_m|} \sum_{k \in \mathbb{V}_m} \mathbf{C}_{n,k}. \quad (4)$$

This contribution is evaluated at the start of training and at predefined epoch intervals. Our approach differs from Fan et al. (2024) in that we use a percentile-based threshold to determine which points to discard. The points with contributions equal to or lower than this bound, including those redundant and never-observed points, will be automatically removed. Tab. 2 validates that our pipeline saves 50% storage and 40% memory, while decreasing time cost and slightly improving performance.

After merging the Gaussians across different blocks, we implement vectree quantization on 2DGS. We first evaluate each point's contribution across all training data. The least important Gaussians undergo aggressive vector quantization on the SHs. The remaining critical SHs, along with other attributes representing Gaussian shape, rotation, and opacity, are stored in float16 format.

## 4 GEOMETRIC EVALUATION PROTOCOLS

The evaluation protocol for rendering quality is well-established and transferable. We adhere to standard practices by measuring SSIM, PSNR, and LPIPS between renderings and groundtruth. How-

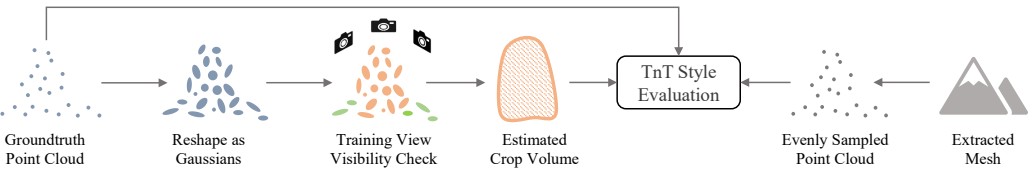

Figure 5: Illustration of the evaluation process.

ever, there is still no universally accepted protocol for assessing geometric accuracy in large-scale scene reconstruction. Recently, *GauU-Scene* (Xiong et al., 2024) introduced the first benchmark, but its evaluation protocol overlooks boundary effects, leading to unreliable assessments. For instance, as indicated in its own paper (Xiong et al., 2024), such a protocol significantly underestimates the geometric accuracy of SuGaR, which demonstrates promising performance in mesh visualization. Moreover, *GauU-Scene* does not align the surface points extraction process across methods, leading to unfair comparison. In particular, NeRF-based methods extract points from depth maps, while 3DGS utilizes Gaussian means. To address these issues, we draw lessons from the evaluation protocol of the Tanks and Temple (TnT) dataset (Knapitsch et al., 2017), which includes point cloud alignment, resampling, volume-bound cropping, and F1 score measurement. For all the compared methods, we first extract mesh and then sample points from the surface. Though TnT's strategy of sampling vertices and face centers is fast, it would underestimate the effect of mistakenly posing large triangles. Therefore, we sample same number of points evenly on the surface.

To further deal with the challenge of boundary effect, an appropriate estimation of the crop volume is necessary. The core here is to check the visible frequency of each point and estimate a bound that can exclude rarely observed points. For efficiency, we take a workaround that formulates points as Gaussian primitives and checks their visibility using a well-optimized GS rasterizer. As illustrated in Fig. 5, we begin by initializing a 3DGS field with the ground-truth point cloud, then traverse all training views to rasterize and count visible frequency through the output visible mask. If the frequency of $j$-th point $\tau_j$ is below a predefined threshold, it will be excluded. Then we calculate the minimum and maximum height of the remaining points, and project them to the ground plane with ground-truth transformation matrix for alpha shape estimation. Given a scene covering $1.47km^2$ with 958 training views and 31.4 million ground-truth points, this process can be completed within 1 minute if rendered in 1080p on a 40G A100. Compared to the crop volume estimated on all points, ours reduces the error bar length of the F1 score from 0.1 to 0.003, enabling a stable, consistent, and reliable evaluation of the model's actual performance.

Aside from automatic crop volume estimation, we also downsample the ground-truth point cloud to accelerate the evaluation process under such large-scale scenes. The downsampling voxel size is set to 0.35m. The distance threshold of $\tau$ varies from 0.3m to 0.6m, according to statistics of nearest-neighbor distances in the downsampled ground-truth point clouds.

## 5 EXPERIMENTS

### 5.1 EXPERIMENTAL SETUP

**Datasets**. We require datasets with accurate ground-truth point clouds. Therefore, we utilize the realistic dataset *GauU-Scene* (Xiong et al., 2024) and the synthetic dataset *MatrixCity* (Li et al., 2023a). From *GauU-Scene*, we selected the Residence, Russian Building, and Modern Building scenes. For *MatrixCity*, we conduct experiments on its aerial view and street view version respectively. Each scene comprises over 4,000 training images and more than 450 test images, presenting significant challenges. These five scenes span areas ranging from 0.3 km$^2$ to 2.7 km$^2$. For aerial views, we follow Kerbl et al. (2023) to downsample the longer side of images to 1,600 pixels. For street views, we retain the original 1,000 × 1,000 resolution. To generate the initial sparse point cloud, we employ COLMAP (Schönberger & Frahm, 2016; Schönberger et al., 2016) along with the provided poses. Ground-truth point clouds are exclusively utilized for geometry evaluation.

**Implementation Details**. All experiments included in this paper are conducted on 8 A100 GPUs. We set the gradient scaling factor $\omega$ to 0.9 and the pruning ratio to 0.025. For depth distortion

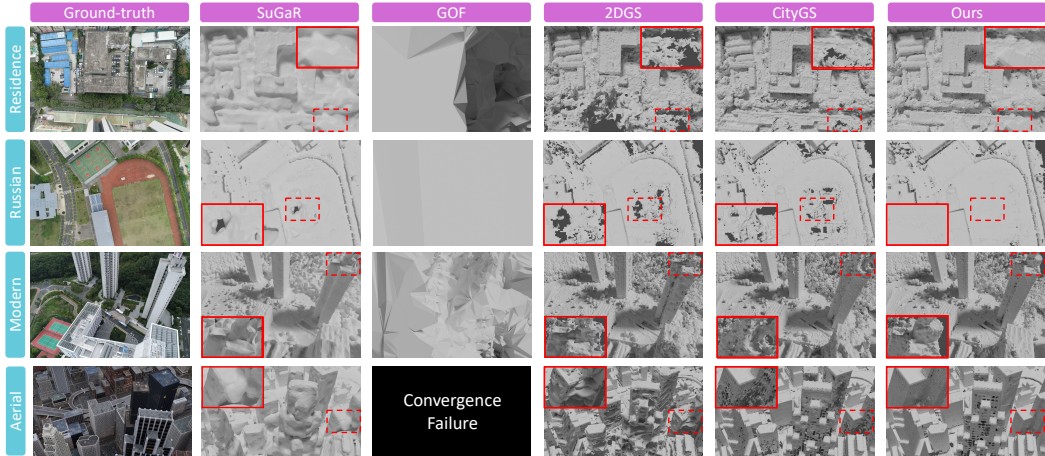

Figure 6: Qualitative comparison of surface reconstruction quality. Here "Russian" and "Modern" denote the Russian Building and Modern Building scene of *GauU-Scene*, respectively. And "Aerial" denotes aerial view of *MatrixCity*. The messy results of GOF are mainly attributed to the near-ground shell-like mesh. More visualizations about GOF and street view scene are in the Appendix.

loss, we empirically find it harmful to performance, and thus set its weight to default value 0. The weight for $\mathcal{L}_{\mathrm{Depth}}$ is exponentially decayed from 0.5 to 0.0025 during both the pretraining and fine-tuning stages. $\mathcal{L}_{\mathrm{Normal}}$ is activated after 7,000 iterations in pretraining and from the beginning in the parallel tuning. Besides, we found that the original normal supervision was overly aggressive for complex scene reconstruction. Consequently, the weight for $\mathcal{L}_{\mathrm{Normal}}$ is reduced to 0.0125, one-fourth of its original value. We adhere to the default settings in CityGaussian (Liu et al., 2024) for the learning rate and densification schedule. Due to page limitations, detailed parameters for block partition and quantization are provided in the Appendix.

For depth rendering, we utilize median depth for improved geometry accuracy, and for mesh extraction, we employ 2DGS's TSDF-based algorithm with a voxel size of 1m and SDF truncation of 4m. Additionally, *GauU-Scene* applies depth truncation of 250m, while *MatrixCity* uses 500m.

**Baselines**. We compare our method against state-of-the-art Gaussian Splatting methods for surface reconstruction, including SuGaR (Guédon & Lepetit, 2024), 2DGS (Huang et al., 2024), and GOF (Yu et al., 2024c). Implicit NeRF-based methods such as NeuS (Wang et al., 2021) and Neuralangelo (Li et al., 2023b) are also included. For a fair comparison, we follow Lin et al. (2024); Liu et al. (2024) to double the total iterations; the starting iteration and interval of densification for GS-based or warm-up and annealing iteration of NeRF-based methods are likewise doubled. We observed that GOF's mesh extraction generates an extremely high-resolution mesh exceeding 1G, significantly larger than the meshes produced by the original settings of SuGaR and 2DGS. To ensure fairness, we adjusted the mesh extraction parameters of these methods to align their resolutions. For large-scale scene reconstruction, we utilize CityGaussian (Liu et al., 2024) as a representative, as other concurrent aerial-view-based methods were not open-sourced at the time of submission. For its mesh extraction, we adopt 2DGS's methodology and use median depth for TSDF integration.

## 5.2 COMPARISON WITH SOTA METHODS

In this section, we compare CityGaussianV2 with state-of-the-art (SOTA) methods both quantitatively and qualitatively. Tab. 1 report results with no compression. As shown, NeRF-based methods are more prone to failure due to the NaN outputs of the MLP or poor convergence under sparse supervision in large-scale scenes. Besides, these methods generally take over 10 hours for training. In contrast, GS-based methods finish training within several hours, while demonstrating stronger performance and generalization abilities. For *GauU-Scene*, our model significantly outperforms existing geometry-specialized methods in rendering quality. As visually illustrated in Fig. 7, our method accurately reconstructs details such as crowded windows and woodlands. Geometrically, our model outperforms 2DGS by 0.01 F1 score. Besides, part (b) of Fig. 1 shows that even without parallel tuning, our proposed optimization strategy enables our model to achieve significantly better

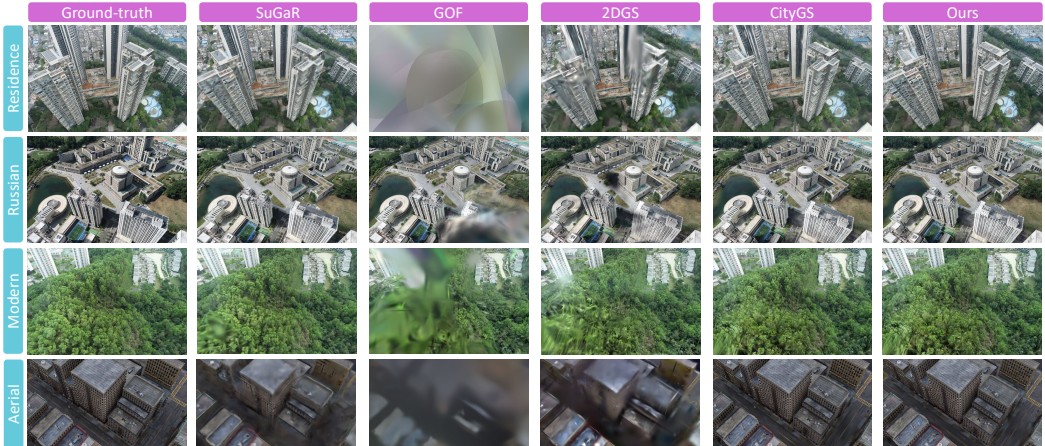

Figure 7: Qualitative comparison of rendering quality. Here "Russian" and "Modern" denote the Russian Building and Modern Building scene of *GauU-Scene*, respectively. "Aerial" denotes the aerial view of *MatrixCity*. The result on street view is included in Fig. 9 of the Appendix.

Table 1: Comparison with SOTA reconstruction methods. "NaN" means no results due to NaN error. "FAIL" means the method fails to extract meaningful mesh due to poor convergence. Here P and R denotes precision and recall against ground-truth point cloud, respectively.

| Methods | GauU-Scene | | | | MatrixCity-Aerial | | | | MatrixCity-Street | | | |
|---|---|---|---|---|---|---|---|---|---|---|---|---|
| | PSNR↑ | P↑ | R↑ | F1↑ | PSNR↑ | P↑ | R↑ | F1↑ | PSNR↑ | P↑ | R↑ | F1↑ |
| NeuS | 14.46 | FAIL | FAIL | FAIL | 16.76 | FAIL | FAIL | FAIL | 12.86 | FAIL | FAIL | FAIL |
| Neuralangelo | NaN | NaN | NaN | NaN | 19.22 | 0.080 | 0.083 | 0.081 | 15.48 | FAIL | FAIL | FAIL |
| SuGaR | 23.47 | 0.570 | 0.292 | 0.377 | 22.41 | 0.182 | 0.157 | 0.169 | 19.82 | 0.053 | 0.111 | 0.071 |
| GOF | 22.33 | 0.370 | 0.390 | 0.374 | 17.42 | FAIL | FAIL | FAIL | 20.32 | 0.219 | 0.473 | 0.300 |
| 2DGS | 23.93 | 0.553 | 0.446 | 0.491 | 21.35 | 0.207 | 0.390 | 0.270 | 21.50 | 0.334 | 0.659 | 0.443 |
| CityGS | 24.75 | 0.522 | 0.405 | 0.453 | 27.46 | 0.362 | 0.637 | 0.462 | 22.98 | 0.283 | 0.689 | 0.401 |
| Ours | 24.51 | 0.576 | 0.450 | 0.501 | 27.23 | 0.441 | 0.752 | 0.556 | 22.24 | 0.376 | 0.759 | 0.503 |

performance in rendering and geometry at both 7K and 30K iterations, while 2DGS struggles to efficiently optimize large and blurry surfels. This validates our superiority in convergence speed. Compared to CityGS, though 0.24 PSNR is sacrificed, our method gains around 11% F1-score improvement. As validated in Fig. 6, our meshes are smoother and more complete.

On the challenging *MatrixCity* dataset, we evaluate performance from both aerial and street views. For MatrixCity-Aerial, our method achieves the best surface quality among all algorithms, with the F1 score being twice that of 2DGS and outperforming CityGaussian by a significant margin. Furthermore, GOF fails to complete training or extract meaningful meshes. In the street view, CityGS and geometry-specialized methods like 2DGS significantly underperform our method in geometry. As illustrated in Fig. 9 in the Appendix, our method provides qualitatively better reconstructions of road and building surfaces, with rendering quality comparable to CityGS.

Regarding training costs, as indicated in Tab. 2, the small version of CityGaussianV2 (ours-s) reduces training time by 25% and memory usage by over 50%, while delivering superior geometric performance and on-par rendering quality with CityGS. The tiny version (ours-t) can even halve the training time. These advantages make our method particularly suitable for scenarios with varying quality and immediacy requirements. Results on other scenes are included in Tab. 4 of Appendix.

## 5.3 ABLATION STUDIES

In this section, we ablate each component of our model design. The upper part of Tab. 2 focuses on the optimization mechanism. As shown, restricting the densification of highly elongated Gaussians has negligible impact on **pretraining** performance. However, as illustrated in Fig. 3, this strategy is

Table 2: Ablation on model components. The experiments are conducted on Residence scene of *GauU-Scene* dataset ((Xiong et al., 2024)). Here we take 2DGS ((Huang et al., 2024)) as our baseline. The upper part ablates on pertaining, while the lower part ablates on fine-tuning. #GS, T, Size, Mem. are the number of Gaussians, total training time with 8 A100, memory, storage cost. *The units are million, minute, Gigabytes, and Gigabytes respectively*. The best performance of each part is in **bold**. "+" means add components on basis of all components in the above rows. An indented line means that only the module in that line is added, while that of other indented rows are excluded. The gray row denotes modification that is aborted and not included in the following experiments.

| Model | Rendering Quality | | | Geometric Quality | | | GS Statistics | | | | |
| | SSIM↑ | PSNR↑ | LPIPS↓ | P↑ | R↑ | F1↑ | #GS | T | Size | Mem. | FPS |
| --- | --- | --- | --- | --- | --- | --- | --- | --- | --- | --- | --- |
| Baseline | 0.637 | 21.12 | 0.401 | 0.474 | 0.362 | 0.410 | 9.54 | **78** | 2.26 | 20.8 | 28.0 |
| + Elongation Filter | 0.636 | 21.18 | 0.401 | 0.477 | 0.362 | 0.411 | **9.36** | **78** | 2.26 | 20.8 | 28.6 |
| + DGD | 0.674 | **22.24** | 0.345 | 0.480 | 0.387 | 0.429 | 9.51 | 84 | **2.25** | 20.7 | 30.3 |
| + Depth Regression | **0.674** | 22.22 | **0.345** | **0.501** | **0.390** | **0.438** | 9.67 | 89 | 2.29 | 25.3 | 29.4 |
| + Parallel Tuning | 0.742 | 23.50 | **0.237** | 0.538 | 0.419 | 0.471 | 19.3 | 195 | 4.57 | 31.5 | 21.3 |
| + Trim (Ours-b) | **0.742** | **23.57** | 0.243 | 0.534 | 0.430 | 0.477 | 8.07 | 179 | 1.90 | 19.0 | 31.3 |
| + Prune | 0.738 | 23.46 | 0.246 | 0.538 | 0.420 | 0.472 | 10.3 | 168 | 1.90 | 24.3 | 30.3 |
| + SH Degree=2 | 0.742 | 23.49 | 0.245 | **0.540** | **0.423** | **0.474** | 8.06 | 176 | 1.29 | 14.2 | 34.5 |
| + VQ (Ours-s) | 0.740 | 23.46 | 0.248 | 0.530 | 0.414 | 0.465 | 8.06 | 181 | 0.44 | 14.2 | 34.5 |
| + 7k pretrain (Ours-t) | 0.721 | 23.17 | 0.281 | 0.517 | 0.416 | 0.461 | **5.31** | **115** | **0.29** | **11.5** | **41.7** |
| + partition of 2DGS | 0.704 | 22.68 | 0.296 | 0.508 | 0.414 | 0.456 | 4.71 | 112 | 0.25 | 11.3 | 43.5 |
| CityGaussian | 0.727 | 23.17 | 0.266 | 0.519 | 0.402 | 0.453 | 8.05 | 235 | 0.44 | 31.5 | 66.7 |

essential for preventing Gaussian count explosion during the **fine-tuning** stage. Additionally, Tab. 2 demonstrates that our Decomposed Densification Gradient (DGD) strategy significantly accelerates convergence, improving 1.0 PSNR, 0.04 SSIM, and almost 0.02 F1 score. A more detailed analysis of how gradient from different losses affects performance is included in the Appendix. The last two lines in the upper section confirm that depth supervision from Depth-Anything-V2 (Yang et al., 2024) enhances geometric quality considerably.

The lower part of Tab. 2 examines our pipeline design. With parallel tuning, both rendering and geometry quality show substantial improvements, validating the success of scaling up. For trimming, we use a more aggressive pruning ratio of 0.1, leading to 50% storage and memory reduction. The result also underscores the importance of trimming for real-time performance. LightGaussian's (Fan et al., 2023) pruning strategy, however, falls short in preserving rendering quality. By using an SH degree of 2 from scratch, we further reduce storage and memory usage by over 25%, with marginal impact on rendering performance or geometry accuracy. And speed is improved by 4.2 FPS. Our contribution-based vectree quantization step takes several minutes for compression, but achieves a 75% reduction in storage. Additionally, by using the result from 7,000 iterations as a pre-train, the total training time decreases from 3 hours to 2 hours, with the model size shrinking to below 300 MB. This compact model is well-suited for deployment on low-end devices like smartphones or VR headsets. However, replacing the block partition with the one generated from 7,000 iterations of 2DGS results in a considerable drop in both the PSNR and F1 score. This suboptimal outcome underscores the importance of fast convergence for efficient training of tiny models.

# 6 CONCLUSION

In this paper, we reveal the challenges of scaling up the GS-based surface reconstruction method and establish the geometry benchmark for large-scale scenes. Our CityGaussianV2 takes 2DGS as primitives, eliminating its problem in convergence speed and scaling up capability. Despite that, we also implement parallel training and compression for 2DGS, realizing considerably lower training cost compared to CityGaussian. Experimental results on multiple challenging datasets demonstrate the efficiency, effectiveness and robustness of our method.

ACKNOWLEDGMENTS

This work was supported in part by the National Key R&D Program of China (No. 2022ZD0116500), the National Natural Science Foundation of China (No. U21B2042, No. 62320106010), and in part by the 2035 Innovation Program of CAS.

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

# A   ADDITIONAL QUALITATIVE COMPARISON

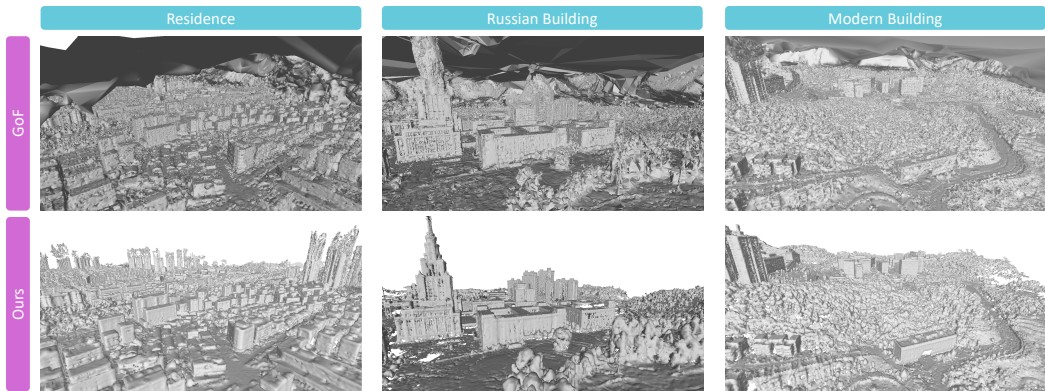

Figure 8: Qualitative comparison of meshes generated from GOF and our CityGaussianV2.

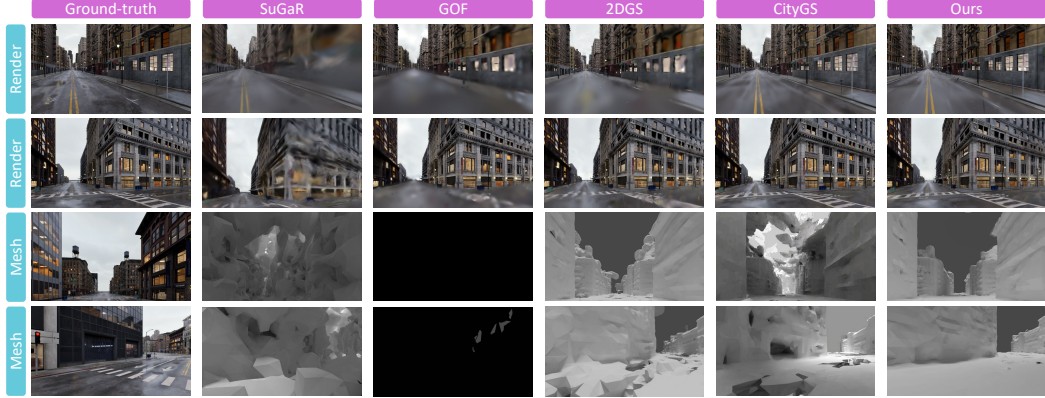

Figure 9: Qualitative comparison of results on the street view of *MatrixCity*.

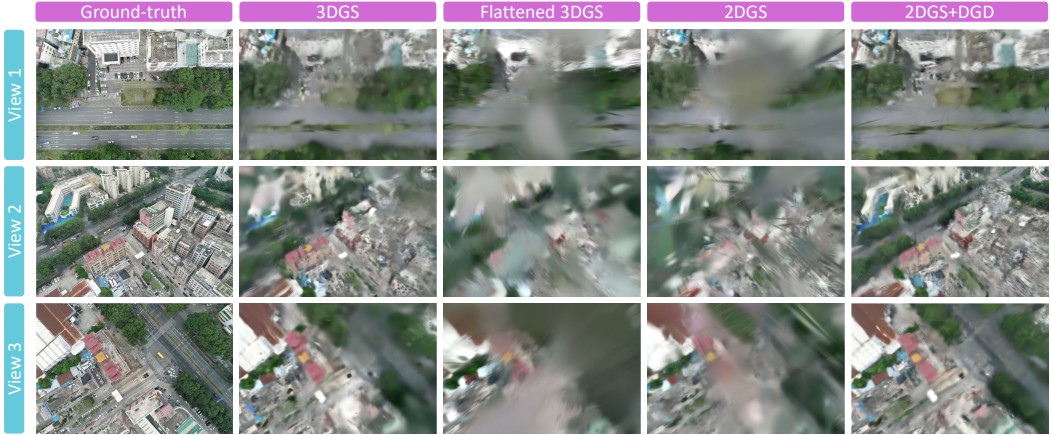

Figure 10: Qualitative ablation of 7K iteration results among different methods.

This section provides additional qualitative comparisons. As illustrated in Fig. 8, the mesh produced by GOF is obscured by a near-ground shell, which obstructs rendering from the test view in Fig. 6 of the main paper and is challenging to remove. However, it does successfully capture the intricate structures of buildings and landscapes. Fig. 8 provides a more thorough comparison. Notably,

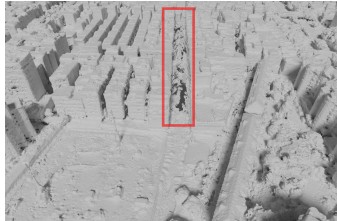 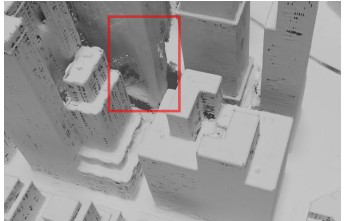 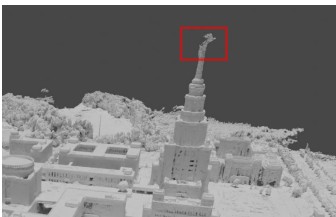

Broken Road Surface      Broken Building Facade      Inaccurate Spire

Figure 11: Bad cases visualization. Due to occlusion and lack of observation, some road surfaces and building facades are not well reconstructed. TSDF-based fusion also struggles to recover some thin structures like spires.

Table 3: Detailed comparison with SOTA on rendering metrics. "NaN" here means no results due to NaN error. "FAIL" means fail to extract meaningful mesh due to poor convergence.

| Methods | GauU-Scene | | | | MatrixCity-Aerial | | | | MatrixCity-Street | | | |
|---|---|---|---|---|---|---|---|---|---|---|---|---|
| | SSIM↑ | PSNR↑ | LPIPS↓ | F1↑ | SSIM↑ | PSNR↑ | LPIPS↓ | F1↑ | SSIM↑ | PSNR↑ | LPIPS↓ | F1↑ |
| NeuS | 0.227 | 14.46 | 0.688 | FAIL | 0.476 | 16.76 | 0.691 | FAIL | 0.562 | 12.86 | 0.514 | FAIL |
| Neuralangelo | NaN | NaN | NaN | NaN | 0.535 | 19.22 | 0.594 | 0.081 | 0.592 | 15.48 | 0.547 | FAIL |
| SuGaR | 0.682 | 23.47 | 0.390 | 0.377 | 0.633 | 22.41 | 0.493 | 0.169 | 0.662 | 19.82 | 0.478 | 0.071 |
| GOF | 0.705 | 22.33 | 0.333 | 0.374 | 0.374 | 17.42 | 0.588 | FAIL | 0.703 | 20.32 | 0.440 | 0.300 |
| 2DGS | 0.756 | 23.93 | 0.232 | 0.491 | 0.632 | 21.35 | 0.562 | 0.270 | 0.723 | 21.50 | 0.477 | 0.441 |
| CityGS | 0.789 | 24.75 | 0.176 | 0.449 | 0.865 | 27.46 | 0.204 | 0.462 | 0.808 | 22.98 | 0.301 | 0.401 |
| Ours | 0.765 | 24.51 | 0.215 | 0.501 | 0.857 | 27.23 | 0.169 | 0.531 | 0.788 | 22.24 | 0.347 | 0.524 |

our CityGaussianV2 showcases qualitatively better reconstructions with more geometry details and fewer outliers.

Fig. 9 visualizes the rendering and extracted mesh of state-of-the-art methods in street view. Our method successfully scales up and the rendering quality is on par with CityGaussian. In terms of geometry, as shown in the last two rows of Fig. 9, the mesh produced by SuGaR appears messy, while GOF is obscured by near-ground shells and rendered in darkness. The road reconstructed by 2DGS is fragmented, and CityGaussian suffers from floating artifacts in the sky. In contrast, our CityGaussianV2 achieves superior quality, constructing a smoother and more complete surface for buildings and roads.

Fig. 10 compares the results at 7,000 iterations across different methods. As shown, 2DGS experiences more severe blurring compared to 3DGS, which significantly hampers its convergence speed. The flattened 3DGS, which modifies 3DGS by constraining one of its scalings to a minimal value as done in Fan et al. (2024), introduces similar blurring effects observed in 2DGS. This suggests that the issue may be inherent to the dimension collapse. In contrast, our DGD strategy leverages the sensitivity of SSIM loss to blurriness, eliminating blurry surfels while enabling much higher quality results at the same 7K iterations.

## B   ADDITIONAL QUANTITATIVE RESULTS

In this section, we present additional quantitative results. Tab. 3 highlights a comprehensive comparison with state-of-the-art (SOTA) methods in rendering metrics. Notably, our approach significantly outperforms geometry-specific methods, while maintaining comparable photometric quality with CityGS. As shown in Fig. 7 of the main paper, our model can achieve qualitatively better reconstructions with more appearance detail.

In addition to the full experimental results in Tab. 1 and Tab. 3, we provide a comparison of parallel training methods with compressed and aligned Gaussian counts in Tab. 4. For reference, we include results where 2DGS is directly paired with CityGS's parallel training strategy. However, as shown, this approach encounters OOM errors in most scenes due to excessive memory demands caused

Table 4: Detailed comparison among SOTA among parallel training methods. 2DGS* here means applying CityGS's training strategy to 2DGS without our proposed optimization mechanism. And "OOM" means one or more sub-blocks fail to finish training due to the out-of-memory error. The best result for specific metrics under each scene is highlighted in **bold**.

| Scene | Method | PSNR↑ | F1↑ | #GS(M)↓ | T(min)↓ | Size(G)↓ | Mem.(G)↓ | FPS↑ |
|-------|--------|-------|-----|---------|---------|----------|----------|------|
| Residence | 2DGS* | OOM | OOM | OOM | OOM | OOM | OOM | OOM |
|  | CityGS | 23.17 | 0.453 | 8.05 | 235 | 0.44 | 31.5 | **66.7** |
|  | Ours | **23.46** | **0.465** | 8.07 | **181** | 0.44 | **14.2** | 45.5 |
| Russia | 2DGS* | OOM | OOM | OOM | OOM | OOM | OOM | OOM |
|  | CityGS | **24.19** | 0.455 | 7.00 | 209 | 0.38 | 27.4 | **55.2** |
|  | Ours | 23.89 | **0.537** | 6.97 | **177** | 0.38 | **15.0** | 33.3 |
| Modern | 2DGS* | OOM | OOM | OOM | OOM | OOM | OOM | OOM |
|  | CityGS | **26.22** | 0.462 | 7.90 | 215 | 0.43 | 29.2 | **57.1** |
|  | Ours | 25.53 | **0.489** | 7.90 | **185** | 0.42 | **16.1** | 34.5 |
| Aerial | 2DGS* | OOM | OOM | OOM | OOM | OOM | OOM | OOM |
|  | CityGS | **27.23** | 0.459 | 10.3 | 217 | 0.56 | 25.7 | **38.6** |
|  | Ours | 26.70 | **0.492** | 10.4 | **181** | 0.56 | **14.8** | 27.0 |
| Street | 2DGS* | **22.24** | 0.371 | 9.20 | 170 | 2.17 | 15.5 | 28.6 |
|  | CityGS | 21.12 | 0.398 | 7.63 | 163 | 0.42 | 11.9 | **50.0** |
|  | Ours | 22.09 | **0.499** | 7.59 | **149** | 0.42 | **10.8** | 31.3 |

Table 5: Detailed geometry metrics on *GauU-Scene* datasets ((Xiong et al., 2024)). * means that the method fails to finish 60,000 iterations training and therefore reports that of 30,000 iterations. "NaN" here means no results due to NaN error, and "FAIL" means fail to extract meaningful mesh.

| Methods | Residence | | | Russian Building | | | Modern Building | | |
|---------|-----------|------|------|------------------|------|------|-----------------|------|------|
|  | P↑ | R↑ | F1↑ | P↑ | R↑ | F1↑ | P↑ | R↑ | F1↑ |
| NeuS | FAIL | FAIL | FAIL | FAIL | FAIL | FAIL | FAIL | FAIL | FAIL |
| Neuralangelo | NaN | NaN | NaN | FAIL | FAIL | FAIL | NaN | NaN | NaN |
| SuGaR | 0.579 | 0.287 | 0.384 | 0.480 | 0.369 | 0.417 | 0.650 | 0.220 | 0.329 |
| GOF | 0.404 | 0.418 | 0.411 | 0.294* | 0.394* | 0.330* | 0.411 | 0.357 | 0.382 |
| 2DGS | 0.526 | 0.406 | 0.458 | 0.544 | 0.519 | 0.531 | 0.588 | 0.413 | 0.485 |
| CityGS | 0.524 | 0.391 | 0.448 | 0.459 | 0.443 | 0.451 | 0.582 | 0.381 | 0.461 |
| Ours | 0.524 | 0.421 | 0.467 | 0.560 | 0.530 | 0.544 | 0.643 | 0.398 | 0.492 |

by redundant Gaussians and the Gaussian count explosion issue illustrated in Fig. 3. Compared to CityGS, our method (the small version, i.e. ours-s in Tab. 2) achieves superior geometric accuracy while significantly reducing training time and memory usage. Under extreme compression (e.g., 75% on Residence) or in street-view scenes, our method also delivers significantly better rendering quality. These results not only highlight the necessity of our proposed optimization strategy but also demonstrate our method's clear advantages over CityGS.

Tab. 5 and Tab. 6 reports detailed performance on *GauU-Scene* dataset. Comparing the quality of the extracted mesh, SuGaR (Guédon & Lepetit, 2024) shows promising precision on the Residence and Modern Building scene, but the overall performance is severely deteriorated by insufficient recall. And GOF (Yu et al., 2024c) fails to finish 60,000 training on the Russian Building scene due to OOM error. 2DGS (Huang et al., 2024) shows competitive geometric performance, substantially outperforming CityGS. However, Tab. 6 showcases that the geometry-specific methods fall short in rendering quality. In contrast, our method not only achieves SOTA surface quality, but also strikes a promising balance with rendering fidelity.

In Tab. 7, we check the influence of different losses in densification. On the one hand, Tab. 7 shows that the most critical gradient for densification is that from L1 RGB loss. Its participation has a negative impact on reconstructing appearance details (SSIM) and overall quality (PSNR). On the other hand, the influence of densification gradient from normal and depth is within the error bar (0.003 for the F1 score). Therefore, we exclusively rely on the gradient from SSIM loss in the official version of our CityGaussianV2.

Table 6: Detailed rendering metrics on *GauU-Scene* datasets ((Xiong et al., 2024)). * means that the method fails to finish 60,000 iterations training and therefore reports that of 30,000 iterations. "NaN" here means no results due to NaN error.

| Methods | Residence | | | Russian Building | | | Modern Building | | |
|---|---|---|---|---|---|---|---|---|---|
| | SSIM↑ | PSNR↑ | LPIPS↓ | SSIM↑ | PSNR↑ | LPIPS↓ | SSIM↑ | PSNR↑ | LPIPS↓ |
| NeuS | 0.244 | 15.16 | 0.674 | 0.202 | 13.65 | 0.694 | 0.236 | 14.58 | 0.694 |
| Neuralangelo | NaN | NaN | NaN | 0.328 | 12.48 | 0.698 | NaN | NaN | NaN |
| SuGaR | 0.612 | 21.95 | 0.452 | 0.738 | 23.62 | 0.332 | 0.700 | 24.92 | 0.381 |
| GOF | 0.652 | 20.68 | 0.391 | 0.713* | 21.30* | 0.322* | 0.749 | 25.01 | 0.286 |
| 2DGS | 0.703 | 22.24 | 0.306 | 0.788 | 23.77 | 0.189 | 0.776 | 25.77 | 0.202 |
| CityGS | 0.763 | 23.59 | 0.204 | 0.808 | 24.37 | 0.163 | 0.796 | 26.29 | 0.160 |
| Ours | 0.742 | 23.57 | 0.243 | 0.784 | 24.12 | 0.196 | 0.770 | 25.84 | 0.207 |

Table 7: Ablation on gradient source of densification. The experiments are conducted on the Residence scene of the *GauU-Scene* dataset ((Xiong et al., 2024)). Here we take 2DGS ((Huang et al., 2024)) with the Elongation Filter as the baseline. #GS and T are the number of Gaussians and total training time with 8 A100 respectively. The best performance of each metric column is highlighted in **bold**. Notably, though the densification future gradients here are not automatically scaled, the numbers of Gaussians are maintained at similar levels.

| Densification Gradient | | | | Rendering Quality | | | Geometric Quality | | | GS Statistics | |
|---|---|---|---|---|---|---|---|---|---|---|---|
| SSIM | RGB | NORM | DEPTH | PSNR | SSIM | LPIPS | P | R | F1 | #GS(M) | T(min) |
| ✓ | ✓ | ✓ | n/a | 0.636 | 21.18 | 0.401 | 0.464 | 0.353 | 0.401 | 9.56 | 78 |
| ✓ | ✓ | | n/a | 0.635 | 21.13 | 0.403 | 0.463 | 0.350 | 0.399 | 9.54 | 85 |
| ✓ | | ✓ | n/a | 0.673 | 22.21 | 0.347 | 0.466 | 0.377 | 0.417 | 9.44 | 85 |
| ✓ | | | n/a | **0.674** | **22.24** | **0.345** | 0.470 | 0.378 | 0.419 | 9.51 | 84 |
| ✓ | | | | **0.674** | 22.22 | **0.345** | **0.490** | **0.381** | **0.429** | 9.67 | 89 |
| ✓ | | | ✓ | **0.674** | 22.21 | 0.347 | **0.490** | 0.380 | 0.428 | 9.43 | 90 |

## C    MORE IMPLEMENTATION DETAILS

For primitives and data partitioning, as well as parallel tuning, we follow the default parameter setting of CityGaussian (Liu et al., 2024) on both aerial view and street view of *MatrixCity* dataset. To be specific, it applies lower learning rates during tuning compared to pertaining, and the street view is trained with a significantly lower learning rate and longer densification interval due to its extreme view sparsity (Zhou et al., 2024). On *GauU-Scene*, we use SSIM threshold $\epsilon$ of 0.05 and default foreground range for contraction, i.e. the central 1/3 area of the scene. The Residence scene of *GauU-Scene* is divided into $4 \times 2$ blocks, while Russian Building and Modern Building scenes are divided into $3 \times 3$ blocks. When fine-tuning on *GauU-Scene*, the learning rate of position is reduced by 60%, while that of scaling is empirically reduced by 20%, as suggested in Liu et al. (2024). For vectree quantization, we set the codebook size to 8192 and the quantization ratio to 0.4.

## D    DISCUSSION

While our method successfully delivers favorable efficiency and accurate geometry reconstruction for large-scale scenes, we also want to discuss its limitations: Firstly, this paper evaluates on the GauUScene and MatrixCity, which feature compensated or ideally constant lighting conditions. Nevertheless, we trust that the consideration of illumination variance and incorporating techniques like decoupled appearance modeling would be helpful for the model's adaptability. Secondly, for mesh extraction, occlusion and lack of observation hinder reconstruction of some road surfaces and building facades. Additionally, TSDF fusion struggles with thin structures, such as spires shown in Fig. 11. Applying more efficient training strategies and advanced mesh extraction algorithms could address these issues. Thirdly, although our compression strategy significantly enhances rendering speed, it still lags behind CityGS even when sharing similar Gaussian counts. Extensive experiments in Tab. 4 validate this conclusion. Future work should explore deeper optimizations of rasterizers,

such as those proposed by Feng et al. (2024), or the integration of Level of Detail (LoD) techniques Kerbl et al. (2024); Ren et al. (2024).

