# OpenReview forum: "CityGaussianV2: Efficient and Geometrically Accurate Reconstruction for Large-Scale Scenes"
_ICLR.cc/2025/Conference — ICLR 2025 Poster_

### Official Review · Reviewer_k1eM · 2024-10-31

**Soundness:** 3
**Presentation:** 3
**Contribution:** 3
**Rating:** 6
**Confidence:** 4

**Summary:**

The paper proposes a large-scale scene geometry reconstruction method based on Gaussian splatting. Specifically, it implements parallel training and compression for 2D Gaussian splatting, incorporates regression based on Depth Anything V2, and designs an Elongation Filter along with a Decomposed-Gradient-based Densification. Additionally, the paper establishes a benchmark for large-scale scene geometry reconstruction for evaluation purposes. The paper demonstrates through comparison experiments that its method achieves superior reconstruction quality on large-scale scenes.

**Strengths:**

1. The paper is clearly written.
2. It implements a comprehensive process for large-scale surface reconstruction, addressing issues in previous methods with thoughtful design, offering a substantial engineering contribution.
3. The paper conducts thorough ablation studies, analyzing the impact of each module on rendering, reconstruction quality, training time, storage requirements, and other factors.
4. The establishment of a large-scale geometry reconstruction benchmark is also a significant and meaningful contribution.

**Weaknesses:**

1. The method comprises multiple combined modules, some of which are known techniques or modified versions of known techniques, so overall, the novelty is somewhat limited. Moreover, the ablation results suggest that some modules have only minor impacts.
2. In the comparison experiments, the authors should also include metrics such as the number of Gaussians, training time, and memory usage for each method, in addition to rendering and reconstruction quality. For certain baseline methods like 2DGS, adjusting hyperparameters can increase the number of Gaussians to enhance the results. The authors should demonstrate that their method has advantages in rendering and reconstruction quality compared to baselines under similar training times and storage usage; otherwise, the results are not fully convincing.

**Questions:**

Apart from the weaknesses mentioned, there are some additional questions:
1. When running the baseline models, was the same hardware used (8 A100 GPUs)? This information should be included in the paper.
2. While the reconstruction quality surpasses CityGS, the rendering quality is slightly inferior across all datasets. What could be the potential reason for this?

---

> ### Author Response · Authors · 2024-11-21
>
> We sincerely thank you for the very constructive comments. We are inspired and hope our discussion brings more insights. If not specified, the referred lines are from the original submission.
>
> # 1. Concern about novelty
> Our method adopts 2DGS primitives with CityGS's model and data division strategy but introduces novel densification and parallel training paradigms. The densification strategy addresses a series of critical challenges preventing 2DGS from trivial scaling up like 3DGS (detailed in Section 3.2, Figures 3 and 10). The latter sets a new end-to-end paradigm that unifies modules in training and compression, enabling a considerable training cost reduction. Both reviewers LweB and M9Gm have acknowledged the difficulty of the problem and the value of our solution. And the insight and novel design of the densification strategy are appreciated by reviewer tjuJ.
>
> # 2. Questions regarding module ablation
> - Key modules—DGD, Depth Regression, and Parallel Tuning—bring significant performance gains.  DGD alone boosts PSNR by 1.1, matching the gain of ScaffoldGS [CVPR’24] over 3DGS on the TnT dataset. Combined with Depth Regression, our method exceeds 2DGS by 0.03 in terms of F1 score, similar to the NeuS [NIPS’21] vs. Geo-NeuS [NIPS’22] difference on TnT. Parallel Tuning further brings a substantial gain of 1.3 PSNR and 0.03 F1.
> - The Elongation Filter doesn't directly improve performance but is critical in preventing Gaussian count explosion during Parallel Tuning (see Figure 3 and experiment in the next question). Modules including Trimming, SH Degree Adjustment, and Vector Quantization reduce storage or memory costs by at least 25%.
> - Therefore, each module plays a vital role, either boosting performance or lowering resource costs.
>
> # 3. Concern about comparison experiments
> - Thanks for your insightful advice. Indeed, the training of existing GS-based surface reconstruction is constrained on single GPU, limiting their capacity. For instance, a single A100 40GB GPU can hold up to 11.2 million Gaussians – well below quality saturation (validated in GrendelGS [arXiv`24]). In contrast, parallel training methods like CityGS and ours can efficiently handle over 20 million points. Given these differences, strict comparability between single-GPU methods and parallel paradigms is challenging.
> - Therefore, we provide a detailed comparison focusing on CityGS, our method, and 2DGS under parallel training (2DGS*). As shown in the table below, 2DGS encounters out-of-memory errors when scaling to most large-scale scenes, highlighting the importance of our optimization. For CityGS and our method, the Gaussian counts are aligned through compression for a fair comparison. The results demonstrate that our method outperforms CityGS in training time, memory cost, and geometric accuracy (F1 score). Furthermore, it shows better rendering quality under extreme compression (75% on Residence) or street view scenes. These results validate the performance advantages of our approach and have been added to the revision. The CityGS's performance in Table 2 has been updated to this aligned version in revision.
> |Scene|Method|PSNR$\uparrow$|F1-Score$\uparrow$|#GS(M)|T(min)|Size(G)|Mem.(G)|FPS|
> |---|---|---|---|---|---|---|---|---|
> |Residence|2DGS*|OOM|OOM|OOM|OOM|OOM|OOM|OOM|
> |Residence|CityGS|23\.17|0.453|8.05|235|0.44|31.5|**66.7**|
> |Residence|Ours|**23.46**|**0.465**|8.07|**181**|0.44|**14.2**|45.5|
> |Russia|2DGS*|OOM|OOM|OOM|OOM|OOM|OOM|OOM|
> |Russia|CityGS|**24.19**|0.455|7.00|209|0.38|27.4|**55.2**|
> |Russia|Ours|23.89|**0.537**|6.97|**177**|0.38|**15.0**|33.3|
> |Modern|2DGS*|OOM|OOM|OOM|OOM|OOM|OOM|OOM|
> |Modern|CityGS|**26.22**|0.462|7.90|215|0.43|29.2|**57.1**|
> |Modern|Ours|25.53|**0.489**|7.90|**185**|0.42|**16.1**|34.5|
> |Aerial|2DGS*|OOM|OOM|OOM|OOM|OOM|OOM|OOM|
> |Aerial|CityGS|**27.23**|0.459|10.3|217|0.56|25.7|**38.6**|
> |Aerial|Ours|26.70|**0.492**|10.4|**181**|0.56|**14.8**|27.0|
> |Street|2DGS*|**22.24**|0.371|9.20|170|2.17|15.5|28.6|
> |Street|CityGS|21.12|0.398|7.63|163|0.42|11.9|**50.0**|
> |Street|Ours|22.09|**0.499**|7.59|**149**|0.42|**10.8**|31.3|
> # 4. Questions about hardware and inferior rendering quality
> - Yes, all experiments were conducted using 8 A100 GPUs. This information has been added to the revised manuscript for clarity.
> - In the absence of additional enhancements,  surface reconstruction methods, as noted in SuGaR, 2DGS, and GOF, typically show lower rendering quality than 3DGS. This is primarily due to two factors: (1) the collapse of primitives from ellipsoids to surfels reduces representation capacity, and (2) geometric constraints restrict the solution space, making it more difficult for the model to find local optimal solution of rendering quality. But it deserves noticing that though in Table 1 our method sacrifices 0.4 PSNR on average, it achieves a significant 0.22 F1-score (18.8%) improvement over CityGS, demonstrating a favorable trade-off for enhanced geometric accuracy.

---

> ### Author Response · Authors · 2024-12-01
>
> Dear Reviewer k1eM,
>
> Thanks again for your insightful comments and valuable time devoted to our paper. As the author-reviewer discussion period is coming to an end, please let us know if there's any further information we can provide to facilitate the discussion process. We are happy to answer any questions or concerns you may have during the author-reviewer discussion period.
>
> We are highly appreciated for your time and consideration.

---

### Official Review · Reviewer_M9Gm · 2024-11-01

**Soundness:** 3
**Presentation:** 4
**Contribution:** 3
**Rating:** 8
**Confidence:** 5

**Summary:**

The authors propose an optimization framework for Gaussian splitting, to address scalability issues when reconstructing large-scale scenes. The proposed framework builds on top of CityGaussian and is named CityGaussianV2 but addresses convergence and scalability issues. They propose the use of DGB densification and depth regression to improve blurry artifacts and accelerate convergence, an elongation filter to mitigate the Gaussian count explosion in 2DGS and a parallel training technique to improve training times and memory consumption.

**Strengths:**

The authors address critical issues in adapting Gaussian splitting techniques in large-scale 3D reconstructions. The proposed framework addressed issues related to scalability as well as reducing the costs associated with training large-scale Gaussian splitting models. The results of the proposed model show that the method is able to scale up CityGaussian without degrading visual quality and geometric accuracy. The paper is well-written and the ideas are well explained. The experimental evaluation is complete and supports the claims in the paper. The authors also provide ablation studies to support their design choices.

**Weaknesses:**

Minor weaknesses overall
1) The novelty of the paper is limited but addresses a very important aspect of large-scale 3D reconstruction using gaussian splitting.

2) Lines 189-190 It looks like R and S symbols are flipped assuming that R is rotation and S is scaling then the decomposition of the covariance matrix makes sense.

3) Line 352-352 How do you estimate the ground plane? Could you provide some more details about the number of points, the scale of the scene, and the hardware used so that this process can be completed in 1 minute?

**Questions:**

See above

---

> ### Author Response · Authors · 2024-11-21
>
> We really appreciate your willingness to recommend our paper for acceptance and attentive review! The responses to the main concerns are as follows:
> # 1. Symbol confusion in lines 189-190
> The error has been fixed in the revision. Thanks for reminding!
> # 2. Details about ground plane estimation in line 352-352
> - The ground plane is estimated using the ground-truth transformation matrix, which aligns the scene's vertical direction with the z-axis. By retaining only the x and y coordinates, we project the points onto the ground plane and calculate the corresponding alpha shape. These details are clarified in the revised manuscript.
> - If the ground-truth matrix is unavailable, the vertical direction can be approximated using PCA on the point cloud or camera poses, enabling an approximate transformation matrix and making our crop volume estimation algorithm applicable.
> - The example processes 31.4 million points with 958 training images (1080p rendering) on a single A100 GPU (40GB), covering 1.47 km², and thus is able to complete within 1 minute. For low-end devices, similar speed can be achieved by lowering the resolution. These details have been added and highlighted to the revision.

---

### Official Review · Reviewer_tjuJ · 2024-11-03

**Soundness:** 3
**Presentation:** 3
**Contribution:** 3
**Rating:** 6
**Confidence:** 4

**Summary:**

This paper presents CityGaussianV2, a novel methodology for 3D scene reconstruction tailored for large-scale urban environments. It builds on the 2D Gaussian Splatting technique by incorporating advanced features such as Decomposed-Gradient-based Densification, depth supervision, and an Elongation Filter. These enhancements collectively aim to improve geometric accuracy and reduce computational demands.

**Strengths:**

The manuscript is well-constructed, with a logical flow that enhances readability.

The use of technical terminology is appropriate, and complex concepts are elucidated with commendable clarity. The integration of innovative techniques like DGD and the Elongation Filter is particularly noteworthy, as these elements not only speed up training convergence but also refine rendering quality, effectively mitigating the Gaussian number explosion issue often seen in 3D Gaussian Splatting of large-scale scenes.

**Weaknesses:**

1. The manuscript explores outdoor reconstruction, yet it does not fully address the incorporation of techniques for improving robustness against variable lighting conditions and viewing angles. Techniques similar to the decoupled appearance models used in GOF could potentially bolster the system's adaptability.

2. The proposed method employs an Elongation Filter to prevent excessive Gaussian numbers, but it lacks a mechanism for handling cases where splitting is necessary. Will directly limiting the Gaussian split result in some areas being under reconstruction? At the same time, why does the number of Gaussians in the method plus an Elongation Filter in Table 2 exceed the baseline? Shouldn't the filters be limited to the number of Gaussian functions?

3. There appears to be a discrepancy between the Gaussian count reported in Figure 3 and the data presented in Table 2. The number of CityGaussian is less than that of the author's method (w Elongation Filter), it is inconsistent with the results shown in Table 2.

4. The limitations are only briefly touched upon in the appendix. A detailed examination within the main body of the paper would provide a more balanced perspective and foster a clearer understanding of the method's scope and potential constraints.

5. The metrics presented in Table 2 (e.g., number of Gaussians, processing time, memory usage) should specify the units of measurement to aid in interpretation.

**Questions:**

Please see the weaknesses.

---

> ### Author Response · Authors · 2024-11-21
>
> We do appreciate your valuable comments and constructive advice, which helped us a lot to make the paper better. The responses to the main concerns are as follows. If not specified, the referred lines are from the original submission.
>
> # 1. Consideration of light condition
> We sincerely thank you for this very constructive comment. Decoupled appearance modeling is very likely to help us improve the model's adaptability. However, our focus during this manuscript's development was on geometric accuracy and training efficiency. Thus, We chose the GauUScene and MatrixCity datasets, which have compensated or constant lighting conditions. We’ve added this to the Discussion section and plan to implement this approach in our code release.
>
> # 2. Discrepancy of reported Gaussian count
> Sorry for the confusion. In Table 2, the Gaussian count (w Elongation Filter) is reported on the **Residence** scene at the **pretrain** stage, where Gaussians are trained for 30,000 iterations with 2DGS’s original hyperparameters. In contrast, as depicted in Lines250-251 and Lines510-511, Figure 3 reports results on the **Rubble** scene at the **finetuning** stage, where Gaussians are divided into submodules for parallel tuning. And Figure 3 illustrates how a specific block fails to tune and how our method addresses this. As such, the reported Gaussian counts differ between these two contexts. To clarify, we have adjusted the expression and emphasized the training stage in the caption of Figure 3 and Lines510-511 in the revision.
>
> # 3. Concern about Elongation Filter
> -   Your insightful comment on the Gaussian count is correct! Upon re-checking the results, we found that the Gaussian count for 2DGS with the Elongation Filter was mistakenly reported; the correct value is 9.36 million. We fix this in the revision and provide additional validation on all GauUScene scenes in the table below. The result shows Elongation Filter does bring a slight Gaussian count drop, but its impact on rendering and geometry performance is minimal. In the example of Figure 3, the filter reduces Gaussians with extreme elongation from 3.83 million to 0.22 million in the **finetuning** stage, significantly easing memory load.
> | 3w Iter.          | SSIM↑ | PSNR↑ | LPIPS ↓ |P↑    | R↑  | F1↑ | #GS(M)|
> | ----------------- | ------- | --------- | --------- | :---: | :---: | :--------: | :--------: |
> | Residence, w/o Elongation Filter  | 0.637 | 21.12 | 0.401| 0.474| 0.362| 0.410| 9.54 |
> | Residence, w Elongation Filter   | 0.636 | 21.18 | 0.401| 0.477| 0.362| 0.411| 9.36 |
> | Russia, w/o Elongation Filter   | 0.767 | 23.23 | 0.220| 0.497| 0.493| 0.495| 11.5 |
> | Russia, w Elongation Filter   | 0.766 | 23.28 | 0.220| 0.493| 0.491 | 0.493| 10.4 |
> | Urban, w/o Elongation Filter   | 0.752 | 25.20  | 0.248| 0.564| 0.395| 0.465| 9.81 |
> | Urban, w Elongation Filter   | 0.750  | 25.16 | 0.249| 0.560| 0.396| 0.464| 9.45 |
> - Considering underreconstruction, Gaussians with extreme elongation under-reconstructed areas still receive gradients, enabling expansion. Since the projected area is more sensitive to the changes of the shorter axis, it expands faster than the longer axis. When the elongation ratio exceeds a threshold, cloning and splitting can still occur, preventing the model from being stuck in under-reconstructed states. Additionally, our tests across various scenes show that extremely elongated Gaussians typically comprise less than 0.5% of all Gaussians. This leaves sufficient opportunities for neighboring Gaussians to fill in under-reconstructed areas. Empirical testing in the table above validates that the Elongation Filter has minimal impact on performance. These findings make us trust that the Gaussians, even with the Elongation Filter, have enough flexibility and redundancy to handle under-reconstruction.
>
> # 4. Illustration of the paper's limitation
> As suggested, a detailed discussion with an analysis of bad cases and extensive examination has been added to the main paper in revision. This enables a clearer understanding of the method's scope and potential constraints. Please refer to Page 10 of the revised version for more details.
>
> # 5. The unit of metrics in Table 2
>  Thank you for your thoughtful comment. The units for the metrics in Table 2 are indeed specified in the corresponding caption of submission. To enhance clarity, we have italicized the unit specifications in the revised version.

---

> ### Author Response · Authors · 2024-12-01
>
> Dear Reviewer tjuJ,
>
> Thanks again for your insightful comments and valuable time devoted to our paper. As the author-reviewer discussion period is coming to an end, please let us know if there's any further information we can provide to facilitate the discussion process. We are happy to answer any questions or concerns you may have during the author-reviewer discussion period.
>
> We are highly appreciated for your time and consideration.

---

> > ### Comment · Reviewer_tjuJ · 2024-12-03
> >
> > Thank you for your detailed response and the revisions to your manuscript based on the feedback. I greatly appreciate your efforts to address the concerns of adaptability to variable lighting conditions and Gaussian count differences.
> >
> > These revisions certainly enhance manuscript address most of my inital concerns. My score has been improved.

---

> > > ### Author Response · Authors · 2024-12-03
> > >
> > > We're glad our response addressed your questions. Thank you again for reviewing our work and providing invaluable suggestions!

---

### Official Review · Reviewer_LweB · 2024-11-04

**Soundness:** 2
**Presentation:** 3
**Contribution:** 2
**Rating:** 6
**Confidence:** 4

**Summary:**

This paper presents CityGaussianV2, a new method for large-scale 3D scene reconstruction that builds on a variant of Gaussian Splatting, 2D Gaussian Splatting.
CityGaussianV2 introduces 3 contributions to resolve scaling problem in large scale scene reconstruction. 1) Depth supervision which uses pre-trained depth estimator to supervise inverse depth, 2) decomposed-gradient-based densification (DGD) which adaptively densifies based on SSIM, and 3) elongation filter to accelerate convergence and mitigate the Gaussian count explosion problem.
In addition to the core technical contributions, the paper presents an optimized parallel training pipeline and new evaluation protocol for large-scale scenes that addresses the limitations of existing benchmarks by accounting for boundary effects.

**Strengths:**

The paper solves one of the tougher remaining challenges in 3D reconstruction with Gaussian Splatting: large scale scene reconstruction. The qualitative and quantitative results of the paper are outstanding. Compared to existing baselines, the paper demonstrates faster convergence, more accurate photo-realistic reconstruction and lower memory usage.
Moreover, the reviewer appreciates additional contributions such as new evaluation criteria that better captures geometric details of larger scenes. The paper spends a lot of effort on being fair on evaluation. The method compares with all SotA methods that the reviewer is aware of.

**Weaknesses:**

The technical novelty of the paper seems rather weak. The paper is built on top of existing works: 2D GS, Depth anything. While the paper has demonstrated a good way to fuse these existing works in a nice scalable manner, the provided contributions seem like multiple ad-hoc solutions to circumvent an underlying core problem: solving densification in a scalable way.

In other words, the reviewer expects solid theoretical reasoning on why their optimization techniques are justified, instead of relying on empirical demonstration of good qualitative results. Otherwise, it is hard to justify whether the contributions of the work provided by authors are coming from meta-fine tuning of the existing 3D reconstruction dataset, or whether they are actual improvements.

**Questions:**

How were the baseline hyper-parameters tuned? It seems to me that the OOM and convergence may be overcome by the best set of hyperparameters in few of the existing models.

How did the authors design the new techniques? I’m curious as to what led to the decisions of such new optimization techniques that are helpful, instead of other potential choices that may be similar in results.

---

> ### Author Response · Authors · 2024-11-21
>
> We are grateful for your insightful and constructive suggestions! The responses to the main concerns are as follows. If not specified, the referred lines are from the original submission.
>
> To begin with, we would like to re-emphasize why achieving geometrically accurate large-scale scene reconstruction is both challenging and non-trivial. While many surface reconstruction methods exist, scaling them to large-scale scenes presents significant obstacles. SuGaR suffers from the limited capacity (Figure 7) and encounters OOM when training on thousands of images due to suboptimal implementation (elaborated in response to Question 2). GOF suffers from severe underfitting, primarily due to large, blurry floaters. These floaters obstruct the field of view and hinder valid supervision (Table 1, Figure 7).  While 2DGS excels in generalization, it faces slow convergence due to blurry floaters (Figures 1, 10) and Gaussian count explosion, leading to OOM (Figure 3). The OOM problem is so severe that 2DGS fails to finish parallel tuning on most scenes (Table 4 in revision). These issues highlight that scaling 2DGS is significantly more challenging than scaling 3DGS, urging the community to carefully consider surface reconstruction method design.
>
> # 1. Concern about Motivation and Novelty
>   The aforementioned challenges strongly motivate our work, and we aim for a solution that is as simple as possible. Our method adopts 2DGS primitives with CityGS's data and model division strategy, but introduces novel densification mechanisms and parallel training paradigms. The proposed densification strategy effectively addresses the critical challenges outlined above, yet it is simple enough to implement with changes of several lines in 2DGS. Furthermore, our training pipeline sets a new end-to-end paradigm that unifies modules in training and compression, enabling a considerable training cost reduction.  Regarding the densification mechanism:
> -  The design of Elongation Filter is based on theoretical analysis in Lines250-262:
>
> > Typically, a 2D Gaussian can collapse to a very small point when projected from a distance, especially those exhibiting extreme elongation [2DGS, Huang et al., 2024]. With high opacity, the movement of these minuscule points can cause significant pixel changes in complex scenes, leading to pronounced position gradients. As evidenced in the left portion of Figure 3, these tiny, sand-like projected points contribute substantially to points with high gradients. And they belong to those with extreme elongation. Moreover, some points project smaller than one pixel, resulting in their covariance being replaced by a fixed value through the antialiased low-pass filter. Consequently, these points cannot properly adjust their scaling and rotation with valid gradients. In block-wise parallel tuning, the views assigned to each block are much less than the total. These distant views are therefore frequently observed, causing the gradients of degenerated points to accumulate rapidly. These points consequently trigger exponential increases in Gaussian count and ultimately lead to out-of-memory errors, as demonstrated in the right portion of Figure 3.
>
> as well as experimental examination exampled in Figure 3. As the Gaussian count explosion stems from extreme surfels, we use a straightforward solution to stop over-cloning: use elongation rate as the indicator to find these surfels and limit their densification.
>   - The Decomposed-Gradient-Densification mechanism is rooted in a thorough analysis of suboptimal convergence. We examine the rendering quality evolution at 1k iteration intervals across 3DGS, flattened 3DGS, and 2DGS, as illustrated in Figure 10. The results indicate that the rendering quality of both flattened 3DGS and 2DGS suffers from blurry floaters that are not split in time. This result also suggests that the problem is inherent to primitive dimension collapse. Since the densification criteria:
> $$
> \begin{aligned}
> Clone: |\partial L/\partial \mu| > \tau_{loss} \land s_{max} < \tau_{radius}; \quad Split: |\partial L/\partial \mu| > \tau_{loss} \land s_{max} > \tau_{radius};
> \end{aligned}
> $$
> is dominated by loss design, the simplest way to eliminate blurriness is to adopt an appropriate gradient source. Theoretically, as detailed in Lines 280-283, SSIM loss is more sensitive to blurriness and structural difference and L1, making it suitable to solve the problem.

---

> ### Author Response · Authors · 2024-11-22
>
> # 2. Solvement of OOM and Convergence problem of baselines
> - For the OOM issue of SuGaR, the primary cause lies in its suboptimal implementation, such as retaining costful but unnecessary gradients during [initialization](https://github.com/Anttwo/SuGaR/blob/7c10c4ae4a267dece512f5c7f40ed212a0a2ab44/sugar_scene/sugar_model.py#L40). Although optimizing the baseline code is beyond the scope of our work, we eliminated these inefficiencies, allowing SuGaR to run successfully on MatrixCity-Aerial. The optimization led to a competitive PSNR of 22.41 and F1 score of 0.169. However, the geometry quality remains inferior to 2DGS. Both qualitative and quantitative results have been updated in the revised version.
> - For the convergence failure in Table 1, we have conducted an extensive search for optimal hyperparameter combinations. Despite these efforts, the improvements were marginal. For example, reducing the scaling learning rate by fourfold upon settings in Lines410-420 improved GOF's PSNR on MatrixCity-Aerial by 0.5. But the final PSNR of 17.94—still 5 points below SuGaR—indicates significant underfitting, which prevents generating meaningful mesh. As previously discussed, the root cause of this underfitting is the presence of large, blurry floaters. Testing the same scene with MipSplatting (the foundation of GOF) produced similar issues, suggesting that the underfitting stems from the inherent design of MipSplatting rather than from hyperparameter settings.

---

> > ### Comment · Reviewer_LweB · 2024-11-27
> >
> > Thank you authors, on thoughtful reply on the comment.
> >
> > Based on the comment provided, the reviewer partially takes back on the comment on the novelty; the reviewer agrees that the problem is challenging, and the solution provided by the authors are one of the workable implementations that deals with the problem. However, reviewer still finds more theoretically established solution over ad-hoc solution to be favorable way to address the scalability problem. For instance, recently published work on spectral-entropy based normalization (Spectral-GS) deals with the same problem which is quite similar to the author's elongation filter, yet provides more thorough theoretical justification. However, Spectral-GS has been published to Arxiv very recently, and should not be considered to be as a target for comparison. Hence, the reviewer finds the issue to be sufficiently addressed for this conference submission.
> >
> > Moreover, on the scalability of SuGaR, thank you for providing more details. It seems like that the SuGaR has sub-optimal quality as they use 3D GS as an engine, instead of surface-friendly 2D GS. It would have been nicer to include SuGaR with 2D-GS backbone as a fair comparison on quality as well as scalability. However, because the authors provided updated numbers, the reviewer finds it to be sufficient for a conference submission.
> >
> > Thus, I'm inclined to changing my rating from marginally below threshold to marginally above threshold.

---

> > > ### Author Response · Authors · 2024-11-27
> > > **Thanks for your thoughtful feedback and for raising your score!**
> > >
> > > Thanks for your thoughtful feedback and for raising your score! We truly appreciate your recognition of the challenges addressed in our work and are grateful for the opportunity to clarify our approach. With your suggestions in mind, we will certainly make more effort to resolve the remaining problems in future works.  Thank you again for reviewing our work and providing thoughtful feedback.

---

### Author Response · Authors · 2024-11-27

Thanks again for your insightful comments and valuable time devoted to our paper. As the author-reviewer discussion period is coming to an end, please let us know if there's any further information we can provide to facilitate the discussion process. We are happy to answer any questions or concerns you may have during the author-reviewer discussion period.

---

### Meta-Review · Area_Chair_mtBn · 2024-12-20

**Metareview:**

This paper presents an optimization framework for Gaussian splatting for addressing the scalability issue in reconstructing large-scale scenes. Built upon the top of CityGaussian, it improves the convergence and scalability issues by the Decomposed-Gradient based Densification (DGD) and Elongation filter. The major strength of the paper is a careful implementation with thoughtful design for realizing the Gaussian splatting for large-scale scenes. The weakness is that the work comprises multiple combined modules that are either known or modified, thus the novelty is somewhat limited. There was an extensive discussion among reviewers and authors particularly about the novelty aspect, and it was agreed that this work is not a straightforward integration of the existing knowledge and agreed that it is a carefully designed working solution for the challenging problem of Gaussian splatting for large-scale scenes. The AC agreed with the reviewers' positive opinions and reached this recommendation.

**Additional Comments On Reviewer Discussion:**

There were two chief assertions regarding the novelty of the work and experimental results. It was mentioned by reviewers that the novelty of the work was limited due to that it comprises of multiple known or modified modules. The concern was addressed through the reviewer-author discussion phase, and the reviewers agreed that it was not a mere combination of the existing methods but it had a thoughtful design to overcome the challenging problem of Gaussian splatting for large-scale scenes. The concerns regarding the experimental validation was also addressed by the authors during the reviewer-author discussion phase by providing additional tables of evaluation.

---

### Decision · Program_Chairs · 2025-01-22

Accept (Poster)